# Towards Cohesive National Surveys in Pakistan: A Comparative Study of DHS and PSLM

Muhammad Ibrahim[1], Nayab Farman[1], Habib Ur Rehman[1], Mujahid Abdullah[1], Amna Mahnoor Cheema[1], Maira Aamir[1], Azadeh Ahmed[1], Ayesha Khan[2], Adnan Ahmad Khan[1,2]*

1 Research and Development Solutions, Islamabad, Pakistan 2 Akhter Hameed Khan Foundation, Islamabad, Pakistan

* adnan@resdev.org

## Abstract

### Introduction

Effective policymaking relies on high-quality data to understand social contexts, identify target populations, and evaluate interventions. In low and middle-income countries (LMICs), household surveys often fill data gaps, providing insights into social dynamics and policy impacts. In Pakistan, the Pakistan Demographic Health Survey (PDHS) and Pakistan Social and Living Standards Measurement (PSLM) are crucial sources of information. While both surveys cover health and socioeconomic indicators, their methodologies and questionnaires vary, leading to potential discrepancies in data.

### Methods

This paper compares PDHS 2017-18, PSLM 2018-19 (provincial level) and PSLM 2019-20 (district level) using family planning and child immunization modules as examples. Similar indicators under each section are examined for differences using weighted proportion t-test. For family planning, we analyzed PDHS 2017-18 and PSLM 2018-19 because PSLM 2019-20, doesn't have family planning section. For immunization, we analyzed PDHS 2017-18, PSLM 2018-19 and PSLM 2019-20.

### Results

Analysis reveals high concordance in family planning indicators with differences of within two percent. Differences in the rates of BCG which is given at birth are under one percent and for the first dose of pentavalent vaccine are near one percent. However, the differences start diverging thereafter and are near nine percent for dose 3 of the pentavalent vaccine. There is high level of concordance between the results of the provincial and district PSLM surveys conducted one year apart.

**Data availability statement:** The Demographic and Health Survey (DHS) data can be accessed through the DHS Program website at https://dhsprogram.com/Countries/Country-Main.cfm?ctry_id=31. The Pakistan Social and Living Standards Measurement (PSLM) data is available on the Pakistan Bureau of Statistics website at https://www.pbs.gov.pk/content/microdata. Access to these datasets may require registration or adherence to specific terms and conditions as specified by the respective data providers.

**Funding:** The manuscript has been prepared by a multidisciplinary team with expertise in public health, epidemiology, and economics. This research was supported by the Bill & Melinda Gates Foundation [grant number: INV-051108], The funders had no role in study design, data collection and analysis, decision to publish, or preparation of the manuscript.

**Competing interests:** The authors have declared that no competing interests exist

## Conclusion

We describe the differences and relative similarities of the PSLM and PDHS surveys, as means to better incorporate their evidence in policy decisions. Both PSLM and PDHS serve a slightly different niche in that PDHS provides more in depth understanding of family planning whereas PSLM connects many health and social indicators with economic measures and gives granularity at the district level. However, to enhance the confidence of policymakers in both the surveys, we describe their concordance and differences and how they may be used in policy decisions.

## 1. Introduction

Effective evidence based policymaking depends on the availability of high-quality data, which can illuminate social contexts, identify appropriate populations to target with interventions, and enable timely evaluation of policies and programs. Conversely, unreliable data with inaccuracies, incompleteness, or the absence of essential administrative records such as vital statistics and public service outputs, limits the understanding of social issues and the impact of interventions [1,2]. Unreliable data undermines policymakers' ability to gauge the effectiveness of their initiatives, potentially leading to inefficient or futile public sector investments. In many low and middle-income countries (LMIC), where program data are unreliable, household surveys are needed to bridge gaps in program data [3–6], to provide insights into social dynamics and track policy and program impacts over time [7–10].

As a developing country, Pakistan relies on a series of surveys that inform various aspects from health, social issue, economics and more (https://www.pbs.gov.pk/). Of these, the leading ones are the Pakistan Demographic Health Survey (PDHS) and Pakistan Social and Living Standards Measurement (PSLM). The PDHS survey aligns with international Demographic Health Surveys (DHS) indicators (https://dhsprogram.com), while the PSLM is based on the World Bank-supported Living Standards Measurement Study (https://www.worldbank.org/en/programs/lsms/overview).

The PDHS includes separate questionnaires for households, ever-married women, ever-married men, biomarkers, and community. It provides extensive details on health indicators and is conducted at the provincial level, taking place sporadically, with intervals of at least five years [11]. It's sampled at the provincial level and includes around 16,000 married women of reproductive age and 4,000 married men.

On the other hand, the PSLM covers a wider range of indicators beyond health and is conducted concomitantly with the Household Integrated Economic Survey (HIES), focusing on income and expenditures from the same household. The PSLM has alternates between provincial and district levels surveys in biennial cycles [12,13]. The provincial PSLM includes around 25,000 respondents who are asked in depth about health and social indicators. The sample for the district version is around 80,000 but with fewer questions across any domain and has fewer domains.

The Demographic Health Surveys and Social and Living Measurement Surveys are also available in many LMICs. The similarities in sampling and questions between the two surveys make them complementary for various health, social, and economic indicators. However, the extent of this complementarity or key differences has never been examined. Another potential option would be to use the relative strengths of these surveys to better inform about social indicators at national and provincial level and allow modeling to infer some indicators at the district levels. As a first step, we compare the PDHS and PSLM provincial surveys in Pakistan,

using immunization, family planning, and women's decision-making modules as examples. This comparison could serve as a model for these surveys conducted in other LMICs.

## 2. Methodology

### 2.1. Structure of the surveys

PDHS and PSLM (provincial and district) cover a diverse range of modules in their surveys (Table 1).

PDHS primarily focuses on health and demographics indicators, while PSLM includes a broader range of socio-economic indicators, including health, demographics, expenses, income, education, consumption, and wealth [14]. The PDHS and the provincial version of

Table 1. Surveys year and modules.

| PDHS | PSLM Provincial | PSLM District |
|---|---|---|
| 2017–18 | 2018–19 | 2019–20 |
| Housing Characteristics | Household Characteristics | Household Characteristics |
| Household Population | Household Roster | Household Roster |
| | | Migration and Functional Limitation |
| | Education | Education |
| | Employment and Income | Employment and Income |
| Reproduction | | |
| Fertility Preferences | Maternal History | |
| Domestic Violence | Women in Decision Making | |
| Contraception | Family Planning | |
| Pregnancy and Postnatal care | Pre and Post Natal Care | Pre- and Post-natal Care |
| | Pregnancy History | Pregnancy History |
| | Maternity History | |
| Child Immunization | Immunization | Vaccination and Diarrhoea |
| | Diarrhoea | |
| Child Health and Nutrition | | |
| Marriage and Sexual Activity | | |
| Husband's Background and Woman's Work | | |
| HIV/AIDS | | |
| Other Health Issues | | |
| | Household Fortnightly, Monthly and Yearly Expenditures | Selected Durable Items Owned by the Household |
| | Food Insecurity Experience Scale (FIES) | Food Insecurity Experience Scale (FIES) |
| | Information Communication and Technology | Information Communication and Technology |
| | Out of Pocket Health Expenditures | |
| | Agricultural and non-Agricultural Establishments | Assets in Possession |
| | Transfers Received and Paid Out | |
| | | Benefits from Services and Facilities |

the PSLM surveys ask more in-depth questions from households (11,856 and 24,809 households, respectively), while the PSLM district version asks fewer questions from a much larger sample of households (176,790).

All three surveys cover basic demographic, education, and employment questions, along with child immunization, diarrhea among children, household characteristics (for estimating wealth index), pre and post-natal care, basic health service questions, and basic knowledge about tuberculosis (TB), Hepatitis B & C. PDHS uniquely captures child nutrition, weight, and height in their biomarker questionnaire, along with questions on domestic violence and gender roles. In contrast, PSLM surveys uniquely capture comprehensive income data, information on communication technology (ICT), and the food insecurity experience scale (FIES). The provincial PSLM survey covers household expenditure and income in depth to create a proper balance sheet, while the district version only includes a short section on income.

PDHS (2017–18) survey covers four provinces, two regions (AJK and GB), (erstwhile) FATA, and Islamabad. Meanwhile, the PSLM provincial-level survey includes four provinces of Pakistan, with FATA being part of KP. To standardize the analysis, we restricted the regions to four provinces (Punjab, Sindh, KP, and Balochistan) and included FATA in KP in PDHS to make it consistent with PSLM.

Surveys were compared in three stages: (1) briefly comparing modules covered by both surveys, (2) reporting questions addressed on family planning and child immunization in both surveys, and (3) comparing estimates of key indicators for family planning and child immunization.

This study utilized modules on 'Reproduction', 'Contraception', and 'Child Immunization' from PDHS and 'Family Planning', 'Women in Decision Making', and 'Immunization' from PSLM. However, the PSLM district-level survey does not include the 'Family Planning' section; hence, we focused on the 'Immunization' section for differential analysis.

The respondents for both surveys were Married Women of Reproductive Age (MWRAs) aged 15–49 years. The sampling strategy, provided by the Pakistan Bureau of Statistics, used the same methodology and enumeration blocks. Table 2 illustrates the rudiments of the two surveys based on the two modules.

Using the described survey modules, two new variables, 'knowledge of family planning' and 'contraceptive method used,' were created because the questions related to these variables were generic rather than specific to the contraceptive method used. There were minor differences in the categories of 'source of the method used' and 'method type of family planning' for PDHS and PSLM. The categories were combined to address this issue of source difference (Table 3). A similar exercise was performed while dealing with contraceptive methods. The analysis gauged the estimates for current users of modern and traditional methods.

The children's sample in the immunization module was restricted to the age of 12–35 months. The age bracket for each vaccine Childhood TB, BCG, Pentavalent (Diphtheria, Tetanus, Pertussis, Hib Pneumonia and Meningitis, Hepatitis B), Polio (Polio-0, Polio-1, Polio-2, Polio-3 and Inactivated Polio vaccine (IPV)) and Measles-1 were assessed at 12–23 months, except for Measles-2 (marked at 24–35 months). PDHS records immunization data on two parameters, 'yes, on record' and 'yes, on recall' for all vaccines, whereas PSLM includes an additional category, 'yes, through campaign' for Polio and Measles vaccines. The primary objective of the analysis was to obtain the percentage of children immunized in both surveys without deviating from the actual number of vaccinated children; therefore, we did not drop or exclude the 'yes, through campaign' category.

## 2.2. Method of analysis

For the comparison analysis between PDHS and PSLM, we calculated the proportions of indicators using area weights in each survey to reduce bias in estimates. These weights also provide auxiliary data on population-known characteristics, thereby reducing sampling errors

**Table 2. Description of modules.**

| | Modules | Year | Regions/Province | Family Planning (MWRA) | Immunization (Children) | |
|---|---|---|---|---|---|---|
| | | | | Age | Age | Age |
| | | | | 15–49 Years | 12–23 Months | 24–35 Months |
| Number of Respondents | PDHS | 2017–18 | National (Federal regions included) | 15,068 | 1,975 | 1,919 |
| | | | Punjab and Islamabad | 4,511 | 1,093 | 975 |
| | | | Sindh | 2,739 | 432 | 448 |
| | | | Khyber Pakhtunkhwa and FATA | 3,390 | 372 | 390 |
| | | | Balochistan | 1,724 | 78 | 106 |
| | PSLM (P) | 2018–19 | National | 24,024 | 3,850 | 4,380 |
| | | | Punjab and Islamabad | 10,316 | 1,599 | 1,706 |
| | | | Sindh | 5,835 | 899 | 972 |
| | | | Khyber Pakhtunkhwa | 5,230 | 905 | 1,095 |
| | | | Balochistan | 2,643 | 447 | 607 |
| | PSLM (D) | 2019–20 | National | – | 17,674 | 22,491 |
| | | | Punjab | – | 8,684 | 10,184 |
| | | | Sindh | – | 3,186 | 4,367 |
| | | | Khyber Pakhtunkhwa | – | 4,051 | 5,336 |
| | | | Balochistan | – | 1,753 | 2,604 |
| Number of Questions | PDHS | 2017–18 | | 30 | 25 | |
| | PSLM (P) | 2018–19 | | 15 | 10 | |
| | PSLM (D) | 2019–20 | | – | 10 | |
| Number of Regions | PDHS | 2017–18 | | 8 | | |
| | PSLM (P) | 2018–19 | | 4 | | |
| | PSLM (D) | 2019–20 | | 4 | | |

Notes: PSLM (P) is the provincial level survey with smaller level sample compared to PSLM (D) which is district level survey covering larger sample size. The Regions covered by PDHS (2017–18) are 'Punjab, Balochistan, KP, Sindh, AJK, GB, FATA, and ICT. While PSLM 2018–19 covered Punjab, Balochistan, KP, and Sindh. In PSLM 2018–19 and PSLM 2019–20, FATA is included in KP.

[15]. We estimated a simple difference between the indicators, followed by a two-tailed proportions test [16,17]. The null hypothesis stated that the difference between weight-adjusted estimates from PDHS and PSLM was equivalent to zero for each indicator of family planning and child immunization. The generic form of the test is given below:

$$H_0 : p_{(PDHS)} - p_{[1]} = 0.$$
$$H_\alpha : p_{(PDHS)} - p_{[1]} \neq .$$

As we are using secondary data for our analysis, institutional review board (ethics committee) approval was not required. These anonymized datasets are available on online portals.

## 3. Results

The PDHS contains more detailed questions on family planning, including the brand names of contraceptives, side effects, sterilization details, timeline of the last method used, and advice by healthcare workers on method use. In contrast, PSLM includes questions about satisfaction levels with the current family planning method (S1 data). Both PDHS and PSLM provincial-level surveys feature similar, in-depth questions about the child immunization module, whereas the PSLM district level survey only asks whether the child is immunized for the recommended vaccines or not. The subsequent subsections present the results for family planning and child from PDHS and PSLM datasets.

**Table 3. How variables were condensed for analysis.**

| Original Category | New Category | Remarks/Notes |
|---|---|---|
| Source of FP Method | | |
| Govt. Hospital | Government | |
| BHU | | |
| RHC | | |
| Govt. Dispensaries/Clinics | | |
| LHWs/LHVs | | |
| NGO | Private | |
| Private Doctors | | |
| Private Hospital | | |
| Private Clinics/Dispensaries | | |
| Dai | Others | |
| Hakeem | | |
| Stores | | |
| Pharmacy/Chemist | | |
| Friends/Relatives/Spouse | | 'Spouse' option is only available in PSLM |
| Types of FP Method | | |
| Withdrawal | Traditional Methods | |
| Rhythm | | |
| LAM | Others | These methods are not stated in PSLM, which has a category named 'others', therefore these 3 categories from DHS were added in 'Others'. |
| Standard Day Method | | |
| Emergency Contraceptive | | |
| Pills | Pills | 'Sterilization' was omitted since PSLM does not mention the time person was sterilized. |
| IUDs | IUDs | |
| Injections | Injections | |
| Implant | Implant | |
| Condom | Condom | |
| Child Immunization | | All categories are used as is; DHS reports 'Rota Virus', but it wasn't used in the analysis since PSLM does not report it. |

## 3.1. Family planning

There is high concordance between the results of the family planning modules from PDHS and PSLM provincial (which is the only version that asks about FP). For example, 99% of Married Women of Reproductive Age (MWRA) in PDHS and 98% in PSLM have heard of at least one family planning method, and 33% and 34% MWRAs were currently using a family planning method, with 26% and 29% using traditional methods, respectively. Similarly, PDHS shows that 46% use modern methods compared to 47% in the PSLM, with a similar method mix. Differences are < 4% for the 'source of methods' and < 6% for 'who makes the decision to use FP', though statistically significant due to large sample sizes. (Table 4)

The PDHS includes more family planning questions about the side effects, problems faced while using contraception, the number of times the respondent used a method to avoid or delay pregnancy, or 'what year was the sterilization performed'. PSLM does not have those questions.

## 3.2. Child Immunization

Differences between PDHS and PSLM provincial-level surveys are more pronounced for immunization (Table 5). For example, differences are < 1% for BCG, which is given within a

**Table 4. Absolute differences in family planning indicators for PDHS and PSLM.**

| Indicators | PDHS 2017–18 (1) | PSLM 2018–19 (2) | Absolute Difference (1–2) | Percent difference = (1–2)/(1)x100 | P-value |
|---|---|---|---|---|---|
| Have heard of contraceptive method | 98.18 | 99.27 | 1.09 | 1.11 | <0.01 |
| Ever used family planning method? | 44.96 | 42.97 | 1.99 | 4.43 | <0.01 |
| Currently using any family planning method | 33.02 | 33.90 | 0.88 | 2.67 | 0.07 |
| Family planning method currently used: | | | | | |
| Pills | 6.56 | 6.96 | 0.40 | 6.10 | 0.12 |
| IUDs | 8.45 | 6.76 | 1.69 | 20.00 | <0.01 |
| Injections | 9.86 | 12.47 | 2.60 | 26.37 | <0.01 |
| Implant | 1.76 | 2.26 | 0.50 | 28.41 | <0.01 |
| Condom | 36.40 | 33.09 | 3.31 | 9.09 | <0.01 |
| Traditional Methods | 36.10 | 37.99 | 1.89 | 5.24 | <0.01 |
| Other Methods | 0.87 | 0.48 | 0.39 | 44.83 | <0.01 |
| Source of Family Planning Method | | | | | |
| Government | 45.01 | 43.90 | 1.11 | 2.47 | <0.01 |
| NGOs/Private doctors/hospitals | 23.70 | 20.40 | 3.33 | 14.05 | <0.01 |
| Others | 31.30 | 35.75 | 4.45 | 14.22 | <0.01 |
| Decision on Family Planning Method: | | | | | |
| Husband/Wife Jointly | 87.04 | 84.06 | 2.90 | 3.33 | <0.01 |
| Husband Alone | 5.80 | 11.59 | 5.79 | 99.83 | <0.01 |
| Wife Alone | 7.05 | 4.35 | 2.70 | 38.30 | <0.01 |

Notes: Other Methods includes Lactational Amenorrhea Method, Emergency Contraception, and Standard Days Method for PDHS and other for PSLM. The category Government includes all government institutions, and the category Private includes all NGOs and other private institutions for PDHS and PSLM. Category 'Others' for the source of family planning method includes store, pharmacy, Dai, and Hakeem for PDHS, while spouse is additionally added to category Others in PSLM.

week after birth and is most likely to be remembered. The differences expand with increasing age of the child, as measured by the differences with each successive dose of pentavalent and pneumococcal vaccines, starting from around 1% for the first dose to just over 9% for the third dose. The differences are higher for measles and over 18% for IPV, which is given sporadically. Polio vaccine recall is distributed non-systematically, as the vaccine is given monthly during campaigns, leading to difficulty in distinguishing between regular and campaign doses. Overall, these differences lead to greater divergence between PDHS and PSLM for immunization at 24%, a full log order higher than the 2% seen for family planning. The divergence increases as the child grows older.

Compared to PDHS and PSLM provincial surveys, differences between provincial and district PSLM are minor. Most vaccine percentage differences are below 1% without any statistically significant difference. However, there is a statistically significant drop in polio vaccine coverage in the district PSLM 2019–20. The difference between basic immunization and all age-appropriate vaccinations (both including and excluding polio vaccines) is higher and increases from 2018–19 (provincial) to 2019–20 (district) PSLM.

## 4. Discussion

Our analysis indicates that while differences between rates in PSLM and PDHS have closed for family planning, and early immunization, the differences in immunization rates diverge after the 3rd month of the child's age. While each survey serves a different niche, our comparison analysis allows the identification of data gaps that can help make these two distinct but complementary surveys more compatible and, in turn, more usable.

There was a 20-percentage points discrepancy between PDHS 2012 and the concurrent PSLM for contraceptive use. This issue was discussed with the survey team at the Pakistan Bureau of Statistics. Their internal review suggested that the discrepancy may have arisen because the interviewees for PSLM could be anyone in the household, not necessarily the MWRA, as is the case with PDHS. Persons other than the MWRA in the household may give different responses, resulting in misleading results [18]. This has since been rectified and is reflected in narrowing of the gap in family planning indicators between the two surveys in 2017–18. The same is true for early age immunization but begins to diverge after the third months of the child's life, perhaps reflecting recall bias of the mother [19].

Family planning is simpler to recall for a woman to remember since she is the primary actor in FP usage, and the use is a one-off event, as only the last status before the survey is included. However, immunization is more complex in that several vaccines are administered, separately and concomitantly, over the first 24 months of a child's life. This is further complicated if there are more than one child of vaccination age in the household. Additionally, routine polio doses and monthly supplemental campaigns are identical and indistinguishable for parents. This creates several data points for a mother to remember, adding to errors in recall [19]. So many vaccinations may be too much for most mothers to remember. As we found, recall is fully concordant between PDHS and PSLM for BCG (given at birth) and first dose of the Pentavalent vaccine (given at 6 weeks). Thereafter, divergence increases with the increasing age of the child and is the highest for polio vaccine, which continues to be given monthly until a child turns five years old [20]. The quality and internal consistency of the

**Table 5. Absolute Percentage differences in child immunization indicator for PDHS and PSLM.**

| Age when vaccine is to be given | Indicators | PDHS 2017-18 (1) | PSLM 2018-19 (2) | PSLM 2019-20 (3) | Absolute Difference (1 - 2) | p-value | Absolute Difference (2 – 3) | p-value |
|---|---|---|---|---|---|---|---|---|
| Birth | Child BCG | 87.92 | 88.70 | 90.75 | 0.78 | <0.01 | 2.05 | <0.01 |
| 6 weeks | Penta-1 | 86.43 | 87.66 | 89.17 | 1.23 | <0.01 | 1.51 | 0.01 |
| 10 weeks | Penta-2 | 82.82 | 86.07 | 86.53 | 3.25 | <0.01 | 0.46 | 0.45 |
| 14 weeks | Penta-3 | 75.68 | 84.78 | 85.22 | 9.10 | <0.01 | 0.44 | 0.48 |
| 6 weeks | Pneumococcal-1 | 85.38 | 87.11 | 87.63 | 1.73 | <0.01 | 0.52 | 0.37 |
| 10 weeks | Pneumococcal-2 | 81.42 | 85.86 | 85.45 | 4.44 | <0.01 | 0.41 | 0.51 |
| 14 weeks | Pneumococcal-3 | 74.96 | 84.63 | 84.46 | 9.67 | <0.01 | 0.22 | 0.72 |
| Birth | Polio-0 | 83.23 | 97.91 | 93.84 | 14.68 | <0.01 | 4.07 | <0.01 |
| 6 weeks | Polio-1 | 94.82 | 98.49 | 92.85 | 3.68 | <0.01 | 5.64 | <0.01 |
| 10 weeks | Polio-2 | 89.97 | 97.68 | 90.56 | 7.71 | <0.01 | 7.12 | <0.01 |
| 14 weeks | Polio-3 | 86.18 | 96.75 | 88.50 | 10.57 | <0.01 | 8.25 | <0.01 |
| 14 weeks | IPV | 63.84 | 81.85 | 82.59 | 18.01 | <0.01 | 0.74 | 0.27 |
| 9 month | Measles-1 | 73.52 | 79.08 | 83.87 | 5.56 | <0.01 | 4.79 | <0.01 |
| | Basic immunization | 66.06 | 76.45 | 81.41 | 10.39 | <0.01 | 4.96 | <0.01 |
| | All age-appropriate vaccinations | 51.29 | 74.79 | 81.17 | 23.50 | <0.01 | 6.38 | <0.01 |
| | All age-appropriate vaccinations (excluding all Polio vaccines) | 56.13 | 76.33 | 82.08 | 20.20 | <0.01 | 5.75 | <0.01 |
| | Measles 2 (age (months): 24-35) | 66.75 | 79.65 | 78.63 | 12.90 | <0.01 | 1.02 | 0.16 |

Notes: The age bracket of the children is set from 12 to 23 months (12 and 23 inclusive), except for Measles 2 for which age is set at 24–35 months. Basic immunization includes BCG, PENTA-1, PENTA-2, PENTA-3, Polio-1, Polio-2, Polio-3, and Measles-1. While All age-appropriate vaccinations include BCG, PENTA-1, PENTA-2, PENTA-3, Pneumococcal-1, Pneumococcal-2, Pneumococcal-3, Polio-0, Polio-1, Polio-2, Polio-3, IPV, and Measles-1.

PSLM surveys is validated by the relatively similarity in results of the provincial and district surveys conducted one year apart (Table 5).

Global evidence suggests that having a written record a memory aid helps improve recall [21–23]. The situation in Pakistan has markedly improved since 2006, when only 10% households had an immunization card [24] in the PDHS 2017–18 survey 63% of MWRAs had a vaccination card present at the time of the survey [11]. Interestingly, only 24% of mothers reported having one in PSLM 2018–19, suggesting a need emphasize asking for one during training for PSLM surveys [12]. It would also be prudent to allow lessons from the in-depth questioning done for immunization in the PDHS and replicate it for the PSLM, as it was done for family planning. Internationally, researchers compare Demographic and Health Surveys (DHS) and Multiple Indicator Cluster Surveys (MICS) to assess health indicators, often identifying discrepancies in postnatal care estimates. MICS provides a more detailed postnatal care module, distinguishing immediate and later care, whereas DHS uses a blended module that does not systematically differentiate these phases [25]. Similarly, discrepancies in immunization estimates may arise because PDHS includes more detailed and specific immunization questions, potentially aiding respondent recall and improving data accuracy.

An additional factor that may contribute to the difference between the vaccination rates between the two surveys is how responses categories for this question are defined in each survey. In PSLM, both provincial and district level, the response category is "Yes, through the polio campaign", while PDHS does not have this category. Table 4 shows that removing the "Yes, through the polio campaign" option from the "all-age appropriate vaccinations" narrows the difference from 23.5% to 20.2%. Thus, the adjustment may have a minor contribution to the difference between the two surveys regarding immunization. Similarly, removing the 'yes, through the campaign' category slightly reduces the difference for Polio and Measles indicators.

The one-year difference between the timing of the surveys may also have accounted for some of the observed differences in the vaccine administration estimates. PDHS took place in 2017–2018, while PSLM provincial surveys took place in 2018–19. For example, in response to a measles epidemic during 2017–2018, a nationwide Supplementary Immunization Activity (SIA) was conducted in October 2018, during which 37.1 million children aged under 5 were vaccinated. An independent post–SIA coverage survey estimated that SIA coverage of 93.3% [10]. The SIA may have also contributed to the difference between the Measles-Containing Vaccine 1 (MCV1) coverage rates, as it occurred between the two surveys. PDHS 2017–18 reported a 73.5% MCV1 coverage, whereas PSLM 2018–19 reported a 79.1% coverage. Similarly, there is a difference in Measles-Containing Vaccine 2 (MCV2) coverage, with PDHS 2018–19 reporting 66.75% coverage and PSLM 2018–19 reporting 79.7% coverage.

## 5. Conclusion

Our paper describes a high-level analysis with some suggestions of using available surveys to inform questions of national policy. We show complementarities, similarities in data collected and in quality, as well as some critical differences. PSLM, conducted biennially, with a larger sample size, and includes comprehensive economic details including income, expenditure, transactions, and assets. It is conducted regularly by the Pakistan Bureau of Statistics with government funds. PDHS is more in depth about health, with a fewer other subjects and smaller sample size, and is conducted ad hoc by the federal health ministry (the Ministry of National Health Services, Regulations and Coordination, through its National Institute of Population Studies), with mostly donor funds. There is a need to rationalize how the two surveys are conducted and how their data are used. A key ask has always been the ability inform about

indicators at the district level. Only PSLM district survey does so, but only for a few indicators. It may be possible to add some indicators to it and then use data from either PDHS or PSLM provincial estimation techniques to impute other data to estimate family planning use which is not measured by the district PSLM survey. An alternative may be to incorporate all questions of PDHS into the PSLM provincial version, perhaps at lower frequency, for example around every 6 years. Understanding these similarities and limitations allows policy makers to develop confidence in these surveys and incorporate their evidence into their decisions.

## Supporting information

**S1 File. Human subjects research checklist.**
(DOCX)

**S1 Data. The table below shows the comparison of questions asked in both PDHS PSLM surveys.**
(DOCX)

## Author contributions

**Conceptualization:** Muhammad Ibrahim, Nayab Farman, Adnan Ahmad Khan.

**Data curation:** Muhammad Ibrahim, Nayab Farman, Habib Ur Rehman.

**Formal analysis:** Muhammad Ibrahim, Nayab Farman, Habib Ur Rehman, Mujahid Abdullah.

**Methodology:** Muhammad Ibrahim, Maira Aamir, Adnan Ahmad Khan.

**Supervision:** Muhammad Ibrahim, Ayesha Khan, Adnan Ahmad Khan.

**Validation:** Adnan Ahmad Khan.

**Writing – original draft:** Nayab Farman, Habib Ur Rehman, Mujahid Abdullah, Maira Aamir, Azadeh Ahmed, Adnan Ahmad Khan.

**Writing – review & editing:** Muhammad Ibrahim, Mujahid Abdullah, Amna Mahnoor Cheema, Maira Aamir, Ayesha Khan, Adnan Ahmad Khan.

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
