## [Decision Letter · Decision Letter 0]

26 Dec 2024

PONE-D-24-46925Towards Cohesive National Surveys in Pakistan: A Comparative Study of DHS and PSLMPLOS ONE

Dear Dr. Khan,

Thank you for submitting your manuscript to PLOS ONE. After careful consideration, we feel that it has merit but does not fully meet PLOS ONE’s publication criteria as it currently stands. Therefore, we invite you to submit a revised version of the manuscript that addresses the points raised during the review process.

This is a beautiful piece of paper that needs a few tweaks to improve the readability for the readers. 

We look forward to receiving your revised manuscript.

Kind regards,

Naveed Sadiq, Ph.D.

Academic Editor

PLOS ONE

Journal Requirements:

“The work has been conducted using funds the Bill and Melinda Gates Foundation [grant number: INV-025171]. “

“The work has been conducted using funds the Bill and Melinda Gates Foundation [grant number: INV-025171]. There are no conflicts of interest. “

“The work has been conducted using funds the Bill and Melinda Gates Foundation [grant number: INV-025171]. “

Additional Editor Comments:

This is a beautiful piece of paper that needs a few tweaks to improve the readability for the readers.

Reviewers' comments:

Reviewer's Responses to Questions

**Comments to the Author**

1. Is the manuscript technically sound, and do the data support the conclusions?

Reviewer #1: Yes

Reviewer #2: Yes

2. Has the statistical analysis been performed appropriately and rigorously? 

Reviewer #1: Yes

Reviewer #2: Yes

3. Have the authors made all data underlying the findings in their manuscript fully available?

Reviewer #1: Yes

Reviewer #2: Yes

4. Is the manuscript presented in an intelligible fashion and written in standard English?

Reviewer #1: Yes

Reviewer #2: Yes

5. Review Comments to the Author

Reviewer #1: Having read through the submission, i see the paper as a vital tool for future programming and an addition to the body of knowledge. This paper once accepted will help in decision making in Pakistan and globally.

Reviewer #2: As a reviewer, I find this article to be logically ordered, well-structured and presented succintly and clearly. The similarities and complexities of the various surveys are well-presented. Save from a few spelling omissions e.g. “in depth (sic)”, the paper is written in a scholarly, yet readable and accessible format.

Some proposed revisions are regarding uniformity and consistency in the number of significant figures e.g. in Table 4 which has both 3 and 4 significant figures and both 1 or 2 decimal places. Table 2 is particularly hard to read in a “wide” format, and a long format is suggested, including disaggregation by provinces of the number of respondents. It would also be helpful to include the number of children for which the MWRA respondents reported, especially given that immunization is a major component of the study. Additional proposed additions to the details on the rudiments of the two surveys include mentioning or clarifying the specific reference or recall periods for the indicators.

The authors could also make a good case advocating for efforts to harmonize survey methodologies, and key indicators across these different survey types. While the authors report differences in respondent selection for certain question types and outbreaks before one of the surveys that resulted in differences, there could be more clarification on immunization indicators in children diverging so much between the two questionnaires and the implications of such, given that both surveys are provincially representative and country-wide potentially even-ing out the impact of the regional immunization activities.

Have concordance surveys taken place between DHS and national surveys in other countries? MICS and DHS? A global perspective would be helpful to add in the discussion.

Overall, a solid paper that discusses a critical need.

6. PLOS authors have the option to publish the peer review history of their article (what does this mean? ). If published, this will include your full peer review and any attached files.

**Do you want your identity to be public for this peer review?** For information about this choice, including consent withdrawal, please see our Privacy Policy .

Reviewer #1: **Yes: ** Ezechukwu Ikenna Nwokoma

Reviewer #2: No

---

## [Author Response · Author response to Decision Letter 1]

21 Jan 2025

We thank both reviewers for reviewing our paper and considering it for publication. We have addressed all comments that required revision and uploaded on the portal for your reference.

---

## [Editor Report · Decision Letter 1]

13 Feb 2025

Towards Cohesive National Surveys in Pakistan: A Comparative Study of DHS and PSLM

PONE-D-24-46925R1

Dear Dr. Khan,

We’re pleased to inform you that your manuscript has been judged scientifically suitable for publication and will be formally accepted for publication once it meets all outstanding technical requirements.

Kind regards,

Naveed Sadiq, Ph.D.

Academic Editor

PLOS ONE
---

## [Editor Report · Acceptance letter]

PONE-D-24-46925R1

PLOS ONE

Dear Dr. Khan,

I'm pleased to inform you that your manuscript has been deemed suitable for publication in PLOS ONE. Congratulations! Your manuscript is now being handed over to our production team.

Kind regards,

on behalf of

Dr. Naveed Sadiq

Academic Editor

PLOS ONE
